# Finerenone: From the Mechanism of Action to Clinical Use in Kidney Disease

**DOI:** 10.3390/ph17040418

**Published:** 2024-03-26

**Authors:** Nejc Piko, Sebastjan Bevc, Radovan Hojs, Robert Ekart

**Affiliations:** 1Department of Dialysis, Clinic for Internal Medicine, University Medical Centre Maribor, 2000 Maribor, Slovenia; robert.ekart2@guest.arnes.si; 2Department of Nephrology, Clinic for Internal Medicine, University Medical Centre Maribor, 2000 Maribor, Slovenia; sebastjan.bevc@gmail.com (S.B.); radovan.hojs@guest.arnes.si (R.H.); 3Medical Faculty, University of Maribor, 2000 Maribor, Slovenia

**Keywords:** chronic kidney disease, diabetes, finerenone

## Abstract

Diabetic kidney disease is a frequent microvascular complication of diabetes and is currently the leading cause of chronic kidney disease and end-stage kidney disease worldwide. Although the prevalence of other complications of diabetes is falling, the number of diabetic patients with end-stage kidney disease in need of kidney replacement therapy is rising. In addition, these patients have extremely high cardiovascular risk. It is more than evident that there is a high unmet treatment need in patients with diabetic kidney disease. Finerenone is a novel nonsteroidal mineralocorticoid receptor antagonist used for treating diabetic kidney disease. It has predominant anti-fibrotic and anti-inflammatory effects and exhibits several renal and cardiac protective effects. This review article summarizes the current knowledge and future prospects of finerenone in treating patients with kidney disease.

## 1. The Epidemiology and Global Burden of Diabetic Kidney Disease

Diabetic kidney disease (DKD) is a frequent microvascular complication of diabetes. It is currently one of the leading causes of chronic kidney disease (CKD) and end-stage kidney disease (ESKD) worldwide [1]. The diagnosis of DKD is based on sustained elevation of urinary albumin excretion (urine albumin to creatinine ratio >30 mg/g) and/or a reduction in estimated glomerular filtration rate (eGFR) to <60 mL/min/1.73 m^2^ in a patient with diabetes. Whenever there is concurrent evidence of diabetic retinopathy, the diagnosis of DKD and the possibility of progressive CKD and ESKD are even more likely [2,3].

The global prevalence of type 2 diabetes was approximately 11% in 2021 and is expected to rise to 12% by 2045 [4]. In the following decades, the most significant relative increase is expected in low- to middle-income countries, especially in Africa, and among children, adolescents, and younger patients [5].

DKD develops in nearly half of patients with type 2 diabetes and one-third of patients with type 1 diabetes. It is estimated that diabetic patients have a two times higher risk of developing CKD than non-diabetic patients [6]. Accounting for nearly half of all cases, diabetes is the leading cause of ESKD worldwide [7]. Although the prevalence of other microvascular and macrovascular complications of diabetes is falling, the number of diabetic patients with ESKD in need of kidney replacement therapy is rising [8,9]. Furthermore, the mortality of these patients increased by 103% from 1990 to 2013, primarily due to cardiovascular events [10]. There is a high unmet treatment need in patients with DKD [11].

## 2. The Pathophysiology behind Diabetic Kidney Disease

Hyperglycemia is associated with several metabolic, hemodynamic, proinflammatory, and profibrotic alterations, which ultimately lead to kidney damage onset and progression to kidney failure [12].

Several metabolic pathways are deranged in the hyperglycemic milieu [13]. Production of advanced glycation end products (AGEs) is increased. Non-enzymatic glycation of various tissue constituents, such as proteins, collagen, lipids, and the extracellular matrix (ECM), can lead to the activation of different cell receptors, culminating in the synthesis and release of nuclear factor κB (NFκB) and reactive oxygen species (ROS). These molecules initiate and maintain kidney damage by cell growth and hypertrophy, inflammation, angiogenesis, endothelial dysfunction, and ECM production [14]. Glycation of cytosolic proteins can reduce nitric oxide (NO) bioavailability and provoke oxidative stress, and collagen glycation outside the cells can impact molecular crosslink and lead to ECM dysfunction. The impact of non-enzymatic glycation appears to be receptor- and non-receptor-based and is present inside and outside the cells [15].

Dysfunction of the intraglomerular hemodynamic apparatus has been implicated in the development and progression of DKD [16]. Glomerular hyperfiltration is a cardinal feature of DKD due to the imbalance of the vasoactive humoral factors that control the pre- and postglomerular arteriolar tone. Factors leading to the vasodilatation of afferent arterioles are reduced NO availability, hyperinsulinemia, and increased production of prostanoids. Increased efferent arteriole resistance is caused by angiotensin II, thromboxane 2, endothelin-1, and ROS [17]. Hyperglycemia is associated with increased glucose reabsorption in the proximal tubule through the upregulation of sodium–glucose cotransporters-2 (SGLT-2). The delivery of sodium to the macula densa apparatus in the distal tubule is therefore reduced, leading to increased activation of the renin–angiotensin–aldosterone system, which is a significant component in not only the hemodynamic but also proinflammatory and profibrotic changes observed in DKD [18]. Arterial hypertension, obesity, and sympathetic nervous system activation cause glomerular hyperfiltration through several independent neurohormonal changes [19].

Hyperglycemia activates several intracellular and epigenetic processes that promote kidney injury through inflammation and fibrosis [20]. Podocytes exposed to AGEs increase NFκB-associated upregulation of messenger RNA expression for various proinflammatory molecules, such as several cytokines (interleukin-18 and interleukin-1β) [21]. AGEs and cytokines cause podocyte injury with subsequent proteinuria, secondary tubular injury, and glomerular endothelial injury [22].

Fibrosis is the final common pathway in progressive kidney injury, no matter the underlying cause. It is initiated by epithelial trauma (due to toxins), inflammation, or direct tubular injury, as observed with albuminuria in diabetes [23]. Increasing evidence suggests that the upregulation of aldosterone is pivotal in progressive fibrosis, CKD, and DKD [24]. Increased renin production causes an increase in angiotensin II, which leads to an increase in aldosterone production through the upregulation of aldosterone synthase. Aldosterone binds to mineralocorticoid receptor (MR), and the bound MR migrates into the nucleus, initiating the transcription of target genes [25].

Classical MR expression is observed in the epithelium of the collecting duct, an aldosterone-sensitive distal nephron. Here, increased sodium and water reabsorption and decreased potassium reabsorption occur. Non-classical MR activation occurs in several glomerular cells, such as mesangial cells, podocytes, and endothelial cells. It has been shown that binding of aldosterone to MR results in activation of platelet-derived growth factor receptor, epidermal growth factor receptor, and PI3K/MAPK signaling, thereby promoting the proliferation of kidney fibroblasts. Additionally, aldosterone-induced epithelial–mesenchymal transition and fibronectin production lead to an increase in ECM [26].

Some new insights have been gleaned regarding the pathophysiology of DKD, recognizing the impact of altered gut microbiota and their metabolites (the so-called gut–kidney axis) on the development of progressive kidney disease in diabetic patients [27]. Genetic susceptibility to DKD has also been noted [28]. Figure 1 presents a schematic overview of crucial pathogenetic events in DKD.

The result of the interplay between different pathophysiologic mechanisms includes structural and histological changes in the kidneys. One of the earliest and most characteristic of all glomerular changes in diabetes is a homogenous thickening of the glomerular basement membrane (GBM) [29]. Thickening and stiffening of the GBM lead to albuminuria and reduce the distensibility of the pericapillary wall, facilitating glomerular injury through hemodynamic changes [30]. Mesangial cells increase the production of matrix proteins, and they undergo proliferation and hypertrophy. These changes lead to some of the vital histological modifications in the diabetic kidney, for example, mesangial expansion, mesangiolysis, and, ultimately, characteristic nodular glomerulosclerosis (Kimmelstiel–Wilson nodules). The resulting reduction in capillary surface area contributes to glomerular hypertension, proteinuria, and reduced glomerular filtration [31]. Diabetes is also associated with recruiting activated T-cells and macrophages into the glomerulus and tubulointerstitium. Tubular injury is common and can be viewed as hypertrophy of proximal tubule cells in the early stages of the disease (due to upregulation of SGLT-2 cotransporters in proximal tubules) and as tubular atrophy in the more advanced stages of DKD [32].

## 3. Diabetic Kidney Disease and Cardiovascular Risk

Diabetes and CKD are cardinal risk factors for cardiovascular disease, and the presence of both synergistically increases cardiovascular risk [33]. Studies have shown that patients with CKD are much more likely to die due to cardiovascular causes than to develop ESKD [34]. Nearly 50% of all patients with CKD stages 4 and 5 suffer a fatal cardiovascular event preceding the onset of developing progressive CKD or ESKD [33].

Additionally, diabetes is independently linked to increased incidence of cardiovascular events, heart failure, and cardiovascular mortality compared to patients without diabetes [35]. It appears that the excess mortality in diabetic patients is mostly present in those with accompanying CKD [36].

Given the close-knit physiology of the metabolic, cardiac, and renal systems, it is not surprising that diabetes frequently coexists with cardiovascular disease and CKD [35]. Diabetes leads to increased oxidative stress, AGEs formation, endothelial dysfunction, and hypercoagulability. Furthermore, more than 80% of diabetic patients have other risk factors for atherosclerosis, such as hyperlipidemia and arterial hypertension [37].

Although patients with CKD, particularly those with DKD, frequently have coexisting traditional risk factors for atherosclerosis, these do not fully explain the increased incidence of cardiovascular events and mortality. It is important to emphasize the impact of several other non-traditional atherosclerosis risk factors present in patients with CKD. These include derangements in calcium–phosphate metabolism, volume overload, arterial stiffness, chronic renal anemia, hyperaldosteronism, chronic inflammation, reduced bioavailability of NO, platelet dysfunction, alterations in plasma levels of clotting factors, and mediators of fibrinolysis [34]. When both diabetes and CKD are present together, the risk for thrombotic events increases greatly [38].

Cardiovascular complications in these patients are heterogenous. CKD is associated with coronary artery disease and increased risk of death and nonfatal cardiovascular outcomes after myocardial infarction [39]. Left ventricular hypertrophy, heart failure, diastolic dysfunction, and sudden cardiac death are regularly observed in these patients. The relative risk of ischemic and hemorrhagic stroke is also increased in those with CKD stage 3 or higher, especially in patients with albuminuria [40]. Finally, peripheral artery disease is present in one in four patients with CKD aged 40 years or more [34]. Taken together, multifactorial and complex pathophysiology, structural and functional vessel wall changes, advanced atherosclerosis, and high thrombosis risk contribute to excess mortality in patients with diabetes and CKD [33,34].

In light of the vulnerability of this patient population, patients with CKD are often underdiagnosed and are less likely to receive appropriate cardiovascular disease risk factor modification. Prompt intervention and new treatment strategies are therefore necessary to improve the prognosis of patients with DKD [41].

## 4. The Role of Steroidal Mineralocorticoid Receptor Antagonists

### 4.1. Preclinical Studies

The protective role of MR antagonists (MRA) in rats with DKD was first described in 2001. Miric et al. (2001) found that short-term use of pirfenidone and spironolactone (both steroidal MRAs) reversed cardiac and renal fibrosis and attenuated increased stiffness [42].

Another steroidal MRA, eplerenone, was also protective in diabetic rats. The authors, Guo et al. (2006), found a reduction in albuminuria and a reversal in histological changes (glomerular hypertrophy, mesangial expansion, and tubulointerstitial fibrosis) in rats with type 1 or type 2 diabetes treated with eplerenone. Furthermore, their results showed that the effect of eplerenone was present without an impact on blood pressure or hyperglycemia [43].

In a rat model of advanced CKD and DKD, chronic spironolactone administration ameliorated glomerulosclerosis and reduced albuminuria, renal connective tissue growth factor levels, collagen synthesis, and macrophage infiltration [44]. Spironolactone treatment in diabetic rats was also associated with intrarenal inhibition of the renal–angiotensin–aldosterone system [45].

### 4.2. Clinical Studies

Mehdi et al. (2009) performed a double-blind placebo-controlled clinical trial in which they included 81 patients with either type 1 or type 2 diabetes. All the included patients were undergoing maximally tolerated inhibition of the renin–angiotensin–aldosterone system. They found that patients receiving 25 mg of spironolactone experienced a 34% reduction in albuminuria compared to the placebo group. There was no difference in blood pressure between patients. Furthermore, 51.8% of the patients in the spironolactone group experienced hyperkalemia at least once during the follow-up period (48 weeks) [46].

Brandt-Jacobsen et al. (2021) performed a similar study by design, but they used high-dose eplerenone (100–200 mg daily) instead of spironolactone. Their findings were similar, but there were no registered episodes of hyperkalemia in the MRA group (follow-up period 26 weeks) [47].

In a systematic review by Mavrakanas et al. (2014), the authors included data from eight studies. They included 404 patients with type 1 or type 2 diabetes and albuminuria; all the patients were on maximally tolerated standard-of-care therapy with renin–angiotensin–aldosterone system inhibitors. During their follow-up period (3 months to 1 year), the patients in the MRA group (either spironolactone or eplerenone) had a 23–61% reduction in albuminuria. Three of the eight studies also showed a reduction in blood pressure in the MRA group. The authors noticed an increase in hyperkalemia episodes in the MRA group, but the dropout rate was still relatively low (less than 17%) [48].

It appears that steroidal MRAs are effective in the treatment of DKD but can cause hyperkalemia. Ferreira et al. (2022) performed a pooled analysis of several randomized controlled trials. Their goal was to analyze the safety and efficacy of steroidal MRAs across the spectrum of eGFR. They mainly included patients with heart failure, myocardial infarction, and CKD. A total of 12.700 patients were included; 2.6% had eGFR less than 30 mL/min/1.73 m^2^. The results showed a reduction in cardiovascular deaths and hospitalizations due to heart failure in the MRA group, but the effect was attenuated as eGFR decreased. They also found a significant increase in investigator-reported hyperkalemia and worsening kidney function in the MRA group, potentially limiting their use in patients with CKD [49].

## 5. The Switch from Steroidal to Nonsteroidal Mineralocorticoid Receptor Antagonists

### 5.1. Finerenone—Why Is It Different?

There has been a remarkable rise in basic research and small clinical trials regarding several pathophysiologic conditions independent of the traditional role of aldosterone in sodium/potassium homeostasis since the late 1980s. These studies have shown an effect of aldosterone on collagen synthesis in coronary arteries in rats [50], the expression of MR in the heart [51], and vasculature [52]. These results stimulated additional research in several inflammatory and fibrotic processes, including CKD [53]. Another trigger for additional research on MRAs was hyperkalemia, which was especially problematic in patients who were already receiving the standard therapy of CKD, angiotensin-convertase inhibitors or angiotensin receptor blockers [54]. After the publication of the RALES study in 1999 (The Randomized Aldactone Evaluation Study), which confirmed the benefit of spironolactone use in patients with heart failure [55], an increase in hyperkalemia-associated hospitalizations in patients on spironolactone was noted. Additional information revealed that elderly patients, patients with CKD, and diabetics were more prone to this common side effect of the drug [54].

A cluster of dihydropyridines (DHPs) acting as MRAs in vitro were identified in an ultrahigh-throughput screening program of nearly 1,000,000 compounds, and chemical optimization of these compounds then led to a novel series of heterobicyclic analogs of naphthyridine derivatives [53,56]. After this finding, new compounds without a steroidal molecule backbone have been developed and named nonsteroidal MRAs to improve the risk–benefit profile of MR-based therapy. Multiple compounds, such as finerenone, esaxerenone, and apararenone, have been designed and tested for several indications (arterial hypertension, CKD, and heart failure) [53].

Finerenone is the first novel nonsteroidal MRA developed with high potency and selectivity for the MR. It inhibits the binding of aldosterone and cortisol and reduces the recruitment of transcriptional cofactors in both the bound and not-bound conformational state of MR [57]. Differently from steroidal MRAs, finerenone has equal tissue distribution between the heart and the kidney, a shorter half-life, no active metabolites, higher MR selectivity than spironolactone, and higher receptor binding affinity than eplerenone [58].

Apararenone is a benzoxazinone derivative and is a long-acting, highly selective MRA. Wada et al. (2021) performed a study on the safety and efficacy of patients with DKD. Their results showed that apararenone reduced albuminuria, and also caused a slight decrease in estimated GFR and an increase in serum potassium. Both of these findings were, however, clinically insignificant [59]. Additionally, a study by Okanoue et al. (2021) showed that apararenone improved several potential liver fibrosis markers in patients with non-alcoholic steatohepatitis [60]. Three randomized studies evaluated the effect of esaxerenone on albuminuria in patients with CKD and type 2 diabetes. A dose-dependent reduction in albuminuria was present in all three studies, and clinically significant hyperkalemia was rarely reported (4–9% of patients) [61]. A meta-analysis by Jiang et al. (2022) showed that the use of nonsteroidal MRAs (finerenone, apararenone, and esaxerenone) in patients with type 2 diabetes and CKD reduced albuminuria and systolic blood pressure without an excess risk of serious adverse events. Apararenone was superior in reducing UACR compared to finerenone, finerenone was superior to esaxerenone in alleviating the decline in estimated GFR, and esaxerenone and apararenone were superior to finerenone in decreasing systolic blood pressure. Finerenone had benefits in reducing the incidence of a sustained decrease of 40% in the estimated GFR from baseline and reducing the risk of hospitalization for heart failure [62].

The rest of this review will focus on finerenone due to its leading role and the amount of preclinical and clinical data.

The differences between finerenone and steroidal MRAs (eplerenone and spironolactone) are presented in Figure 2.

### 5.2. Preclinical Data

Gil-Ortega et al. (2020) performed a study on the effect of finerenone on albuminuria and arterial stiffness in a genetic model of CKD Munich Wistar Frömter (MWF) rats. They found that a four-week treatment with finerenone reduced intrinsic arterial stiffness, oxidative stress, and albuminuria [63].

In a similar model, Gonzalez-Blazquez et al. (2018) studied the effects of finerenone on endothelial dysfunction, oxidative stress, and albuminuria. Their results showed that finerenone treatment for four weeks decreased endothelial dysfunction (which was measured by an increase in NO availability and by an increased relaxation of the aorta after stimulation by acetylcholine) and decreased oxidative stress (measured by an increase in total superoxide dismutase activity). Albuminuria dropped by 40% after starting finerenone, and systolic blood pressure was also significantly lower [64].

Cardioprotective and anti-fibrotic properties of finerenone were demonstrated in a study by Grune et al. (2018) in which the authors found that the inhibition of profibrotic MR activation was linked to decreased cardiac fibrosis and improved left ventricular function in a mouse model [65].

Kolkhof et al. (2014) found that finerenone treatment prevented rats from developing functional and structural kidney and heart damage at dosages not reducing systemic blood pressure. In their study, finerenone reduced cardiac hypertrophy, plasma prohormone of brain natriuretic peptide, and proteinuria more efficiently than eplerenone when comparing equinatriuretic doses. In rats that developed chronic heart failure (after ligation of a coronary artery), finerenone but not eplerenone improved systolic and diastolic left ventricular function and reduced plasma prohormone of brain natriuretic peptide levels. They discovered an equal distribution of finerenone in rat cardiac and renal tissues using quantitative whole-body autoradiography. The risk of electrolyte disturbances was low [66].

### 5.3. Clinical Data

Finerenone in Reducing Kidney Failure and Disease Progression in Diabetic Kidney Disease (FIDELIO-DKD) was a phase 3 randomized double-blind placebo-controlled multicenter clinical trial. In this study, authors Bakris et al. (2020) randomly assigned 5734 patients with CKD and type 2 diabetes in a 1:1 ratio to receive either finerenone or placebo. The eligible patients had moderate albuminuria (urinary albumin/creatinine ratio 30–300 mg/g), eGFR 25–60 mL/min/1.73 m^2^, and diabetic retinopathy, or they had severe albuminuria (over 300 mg/g) and eGFR 25–75 mL/min/1.73 m^2^. The patients in the finerenone arm were prescribed either 10 mg (estimated GFR at inclusion more than 25 mL/min/1.73 m^2^ and less than 60 mL/min/1.73 m^2^) or 20 mg (estimated GFR at inclusion 60 mL/min/1.73 m^2^ or more). An increase in the dose from 10 to 20 mg once daily was encouraged after 1 month, provided the serum potassium level was 4.8 mmol/L or less and the estimated GFR was stable; a decrease in the dose from 20 to 10 mg once daily was allowed any time after the initiation of finerenone or placebo. The primary outcomes were renal events (kidney failure, a sustained decrease in eGFR of at least 40% from the baseline value, or death from renal causes). The secondary outcomes were cardiovascular events (death from cardiovascular causes, nonfatal myocardial infarction, nonfatal stroke, or hospitalization for heart failure). During the follow-up period of 2.6 years, a primary event occurred in 17.8% of the finerenone group, compared to 21.1% in the placebo group (hazard ratio 0.82, *p* = 0.001). A secondary event occurred in 13.0% of the finerenone group, compared to 14.8% in the placebo group (hazard ratio 0.86, *p* = 0.03). The incidence of hyperkalemia was higher in the finerenone group compared to the placebo group (2.3% vs. 0.9%). However, in most cases, hyperkalemia was not clinically significant and did not lead to finerenone withdrawal, hospitalizations, or death [67].

Similar by design, the Finerenone in Reducing Cardiovascular Mortality and Morbidity in Diabetic Kidney Disease (FIGARO-DKD) trial was primarily designed to assess the effect of finerenone on cardiovascular outcomes. Seven-thousand-four-hundred-thirty-seven patients underwent randomization in a 1:1 ratio (finerenone 10 mg or 20 mg vs. placebo; median follow-up period 3.4 years). A 13% reduction in cardiovascular events was noted in the finerenone group, mainly due to a reduction in hospitalizations for heart failure (29% decrease). Additionally, the authors, Pitt et al. (2021), found a 13% reduction in renal outcomes. Adverse events were comparable between the two groups, with a slightly higher incidence of hyperkalemia in the finerenone group (1.2% vs. 0.4%). Similar to FIDELIO-DKD, hyperkalemia in FIGARO-DKD was not associated with severe outcomes, such as hospitalizations or death [68].

A pooled secondary analysis was also performed by Agarwal et al. (2022) (FIDELITY—Combined FIDELIO-DKD and FIGARO-DKD Trial programme analysis), combining the patients from the studies above. The results of the FIDELITY analysis confirmed the favorable effects of finerenone on the reduction in cardiovascular risk and kidney disease progression in a broad range of patients with CKD and type 2 diabetes (23% relative risk reduction in kidney composite outcomes; 14% relative risk reduction in cardiovascular composite outcomes). The risk of adverse events, including acute worsening of kidney function and hyperkalemia, was very low [69].

Agarwal et al. (2023) performed a post-hoc analysis of the FIDELITY analysis in which they specifically analyzed the likelihood of hyperkalemia and systolic blood pressure lowering in patients with resistant hypertension and CKD. The difference in systolic blood pressure after 17 weeks was 7.1 mmHg for finerenone vs. 1.3 mmHg for the placebo. The incidence of serum potassium above 5.5 mmol/L was 12% for finerenone vs. 3% for the placebo. The treatment discontinuation for finerenone was 0.3% vs. 0.0% for the placebo [70]. Compared to the AMBER trial (patiromer vs. placebo to enable spironolactone use in patients with resistant hypertension and chronic kidney disease), the likelihood of hyperkalemia was much lower (35% for spironolactone + patiromer vs. 64% for spironolactone without patiromer) [70,71].

Goulooze et al. (2022) designed nonlinear mixed-effects population pharmacokinetic/pharmacodynamic models to analyze the finerenone dose exposure–response for potassium in FIDELIO-DKD. Although the observed potassium levels decreased with increasing dose, model-based simulations revealed that this was due to the potassium limit for inclusion and uptitration of finerenone from 10 to 20 mg in a time-dependent manner (≤4.8 mmol/L after one month of treatment). When proposing a fixed dosing regime, hyperkalemia was more common in those who were taking 20 mg of finerenone daily [72].

Several clinical trials are currently ongoing. The FINE-REAL study (NCT05348733) is a non-interventional observational study aimed to provide insights into the use of finerenone (10 mg and 20 mg) in a real-life clinical setting [73]. The FINE-ONE trial (NCT05901831) is a randomized double-blind placebo-controlled trial on the use and effects of finerenone (10 and 20 mg) in patients with DKD and type 1 diabetes. If successful, finerenone could become the first registered treatment for CKD in type 1 diabetes in almost 30 years [74]. The FIND-CKD study (NCT05047263) is a double-blind randomized multicentric international study aimed at assessing the role of finerenone (10 and 20 mg) in treating albuminuric CKD in patients without diabetes. It is supposed to end in 2026 and could impact the treatment of several other non-diabetes-associated glomerular diseases [75]. The EFFEKTOR trial (NCT06059664) is a vanguard multicenter phase 2 randomized double-blind placebo-controlled clinical trial to determine the feasibility, tolerability, safety, and efficacy of finerenone in kidney transplant recipients. It is supposed to be completed in December 2025. The resume of the most important clinical trials on finerenone in kidney disease is presented in Figure 3 [67,68,69,73,74,75].

Drug–drug interactions with finerenone have been investigated in healthy male volunteers. Finerenone is mainly metabolized by CYP3A4 in the gut wall and liver. Concomitant use of finerenone with strong CYP3A4 inhibitors, such as ketoconazole, ritonavir, and grapefruit juice, can lead to increased exposure to the drug and hyperkalemia. Other drugs than can increase the likelihood of hyperkalemia in patients with finerenone include erythromycin, verapamil, fluvoxamine, potassium-sparing diuretics, angiotensin-convertase inhibitors, angiotensin receptor blockers, and other MRAs, such as spironolactone and eplerenone. The risk of hypotension increases with the use of other antihypertensive agents [76].

While finerenone is more expensive compared to steroidal MRAs, studies have shown that the reduction in renal and cardiovascular events is so significant that it can lead to a decrease in total lifetime costs per patient (compared to standard-of-care therapy), making it a cost-effective option [77].

### 5.4. Holistic Approach to Diabetic Kidney Disease

Lifestyle measures are the basis of diabetes and DKD treatment. They include reducing dietary protein intake, ensuring adequate fruit and vegetables, following a Mediterranean diet, and considering the need for vitamin and mineral supplements. Some individuals have to reduce potassium and fluid intake as well [78].

SGLT-2 inhibitors promote glucosuria, reduce intraglomerular pressure through vasodilatation of the afferent arteriole, and normalize the tubuloglomerular feedback mechanism. They also show anti-inflammatory, anti-fibrotic, and oxidant mechanisms. SGLT-2 inhibitors are cardio- and renoprotective and are currently used as a firstline therapy for diabetes, along with metformin [79].

Glucagon-like peptid-1 receptor agonists (GLP-1RA) are a group of drugs used to treat type 2 diabetes and effectively lower glycated hemoglobin. The activation of GLP-1 receptors increases insulin secretion and reduces beta cell apoptosis and glucagon release in a glucose-dependent way. Furthermore, they favor weight loss in obese patients through an increase in satiety, reducing appetite, delaying gastric emptying, and potentially increasing thermogenesis of brown adipose tissue. They have several direct renoprotective effects, mainly through promoting natriuresis and inhibiting inflammation and fibrosis [80].

Renin–angiotensin–aldosterone system inhibition is pivotal in treating DKD. Cornerstones of treatment for the last 30 years have been and still are angiotensin-convertase inhibitors and angiotensin receptor blockers. Additionally, nonsteroidal MRA finerenone has now been approved by the European Medicines Agency and The United States of America Food and Drug Administration for the treatment of DKD with albuminuria. By using drugs with pleiotropic mechanisms of action (SGLT-2 inhibitors, GLP-1RAs, and finerenone), we are changing the landscape of DKD and moving from solely kidney protection to combined kidney and cardiovascular protection. Hopefully, this will greatly improve short- and long-term outcomes in this patient population [79].

### 5.5. Looking beyond Diabetic Kidney Disease

In a preclinical study by Zhu et al. (2023), the authors found that combined blockade of the renin–angiotensin–aldosterone system, SGLT-2 receptors, and MR resulted in reduced tubulointerstitial inflammation and fibrosis in mice with Alport syndrome, highlighting the potential role of drugs in treating collagenopathies [81]. In a similar preclinical model, Tu et al. (2022) found that treatment with finerenone reversed established pulmonary hypertension, total pulmonary vascular resistance, and vascular remodeling in studied rats. They also found that continued finerenone treatment decreased inflammatory cell infiltration and vascular cell proliferation in the lungs of the studied rats [82].

Mårup et al. (2024) performed a randomized controlled trial on the benefit of each of finerenone and dapagliflozin, as well as their combination, on albuminuria, estimated GFR, and safety parameters in patients (*n* = 20) with non-diabetic CKD. Their results showed a significant decrease in UACR after receiving monotherapy with either finerenone or dapagliflozin, and the effect was even more apparent in patients who received combination therapy. At 8 weeks, systolic blood pressure and estimated GFR were reduced by 10 mmHg and 7 mL/min. Adverse effects were minimal, and neither drug required discontinuation in any participant during the study period. The likely explanation for the benefit of finerenone in this patient population is a reduction in intraglomerular pressure and additional blockade of profibrotic and proinflammatory pathways [83].

In addition to offering profound cardiovascular benefits, finerenone also leads to an independent decrease in new-onset atrial fibrillation in patients with CKD and type 2 diabetes [84].

There are several ongoing clinical trials on the use of finerenone in heart failure patients. The Finerenone in Heart Failure patients (FINEARTS-HF, NCT04435626) trial is a phase III randomized clinical trial aimed at evaluating the efficacy and safety of finerenone in symptomatic heart failure patients (NYHA classes II–IV and left ventricular ejection fraction 40% or more) on top of standard of care therapy [85]. The REDEFINE-HF trial (NCT06008197) will assess the role of finerenone as monotherapy in treating patients with mildly reduced or preserved ejection fraction, and FINALITY-HF (NCT06033950) will assess the role of finer enone as monotherapy in patients with reduced left ventricular ejection fraction (less than 40%). The open-label CONFIRMATION-HF trial (NCT06024746) will study finerenone in combination with SGLT-2 inhibitor versus standard of care in patients recently hospitalized due to heart failure [85]. These studies will further define and expand the role of finerenone across the spectrum of cardiorenal medicine [85].

## 6. Conclusions

Finerenone is a nonsteroidal highly selective MRA with protective renal and cardiovascular benefits, predominantly due to anti-inflammatory and anti-fibrotic effects. It has a favorable safety profile and has a lower tendency to cause hyperkalemia when compared to steroidal MRAs. It is currently used for patients with DKD and albuminuria. There are several ongoing clinical trials that could lead to the expanded use of finerenone across the spectrum of CKD and heart failure.

## Figures and Tables

**Figure 1 pharmaceuticals-17-00418-f001:**
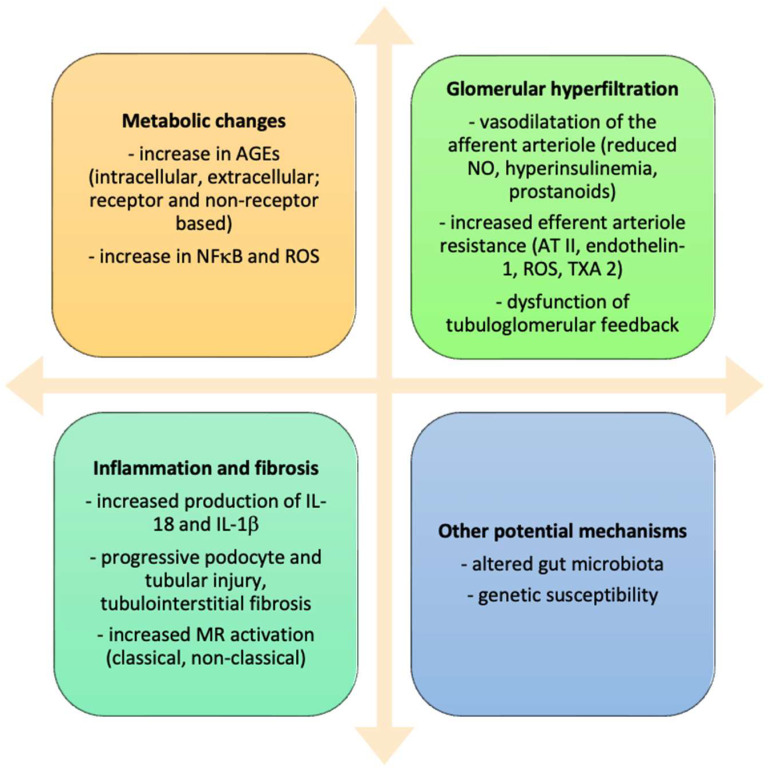
Main components in the development of diabetic kidney disease. Combined, several metabolic changes, glomerular hyperfiltration, inflammation, and fibrosis contribute to the development of diabetic kidney disease. Additional novel mechanisms include altered gut microbiota (gut–kidney axis) and genetic factors. Legend: AGEs—advanced glycation end products; NFκB—nuclear factor κB; ROS—reactive oxygen species; IL-18—interleukin 18; IL-1β—interleukin-1β; MR—mineralocorticoid receptor; NO—nitric oxide; AT II—angiotensin II; TXA 2—thromboxane 2.

**Figure 2 pharmaceuticals-17-00418-f002:**
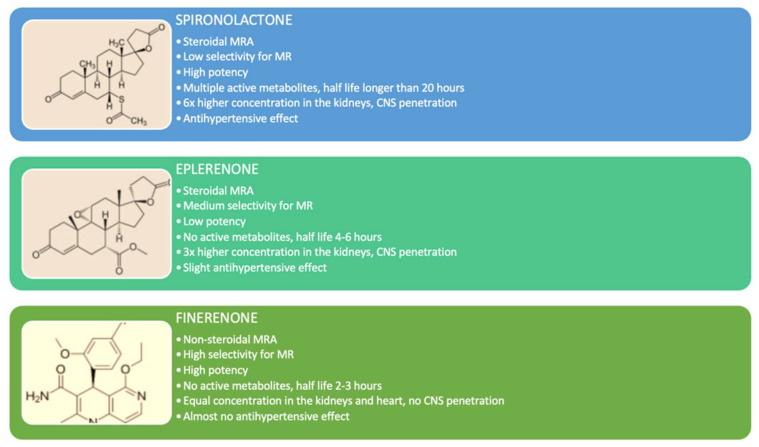
The comparison between steroidal mineralocorticoid receptor antagonists (spironolactone and eplerenone) and finerenone [58]. Legend: MRA—mineralocorticoid receptor antagonist; MR—mineralocorticoid receptor; CNS—central nervous system.

**Figure 3 pharmaceuticals-17-00418-f003:**
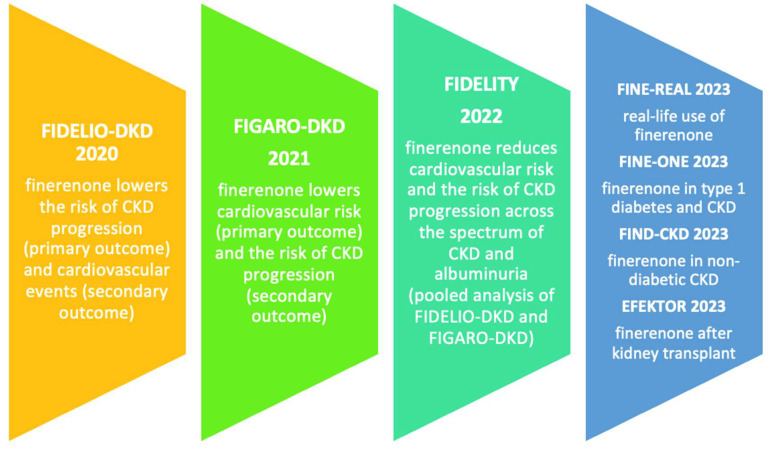
Finerenone in diabetic kidney disease—most critical clinical trials [67,68,69,73,74,75]. Legend: CKD—chronic kidney disease.

## Data Availability

Data sharing is not applicable.

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
