# Peer review of "Finerenone: From the Mechanism of Action to Clinical Use in Kidney Disease"

_pharmaceuticals, 2024, doi:10.3390/ph17040418_

Round 1
Reviewer 1 Report
Comments and Suggestions for Authors
Author Response
Reviewer #1
The present review was overviewed the pathophysiology, clinical features and presentation of diabetic kidney disease and chronic kidney disease, potential therapeutic targets of finerenone. Authors highly motivated and discusses unique ideas, but there were some points to be need to be reconstructions of the present format.
- Page-2; paragraph-2, the second sentence is too long and its very complicated to
understand. There is no clear introduction section written this manuscript, its need to revise.
Response: Thank you for your comment. To make things clearer we have decided to remove the second sentence. It is now apparent that the main focus of this paragraph is to emphasize that the prevalence of type 2 diabetes is increasing, especially in the low and middle income countries.
- After sub-heading 1, the author has to explain more elaborately about the relationship and interconnection between the DKD-cardiovascular disease and CKD.
Response: Thank you for this constructive comment. We have added an additional subheading, including the interconnection between the DKD, CVD and CKD.
- Figure 1 need to revise diagrammatic representation of interconnection between the DKD-cardiovascular disease-CKD and it signaling pathways.
Response: Thank you for your comment. We have added additional explanation and summarization of Figure 1. Also, we have added another subheading regarding the CVD in this population, expanding on the pathophysiology of DKD.
- Similarly, between the sub healing-2 and 3, there is no continuity, need to add more vital points.
Response: Thank you for your valuable comment. We have added more data on cardiovascular disease and the role of prompt intervention and new treatment startegies in this population, in order to improve the continuity of the manuscript.
- The entire manuscript, inside the sentence the authors followed author name alone, better need to include author name with year like “Mehdi et al. 2009”
Response: Thank you for your comment, we have corrected this.
- The history and nomenclature of finerenone need to give introduction before starting its
experimental results.
Response: Thank you for your valuable comment. This has now been corrected, we have added several new paragraphs on the origins of finerenone and pathway to discovery.
- Page -7; the authors introduced the non-steroid MRA such as “finerenone, esaxerenone and apararenone” but in text missing esaxerenone and apararenone explanations, its
comparison with finerenone.
Response: Thank you for your comment. We have added new data and information on esaxerenone and apararenone, as well.
- Later on, its need to describe briefly with finerenone safe dosage, clinical significance and its over dosage mediated hyperkalemia. Need to compare hyperkalemia to discuss with steroid versus non-steroid MRA.
Response: Thank you for this comment. We have added data on dosage and hyperkalemia incidence and prevalence when using finerenone.
- Missing the goal, future prospectus and summary of finerenone in DKD.
Response: Thank you for your comment. We have added a subheading and summary of novel approaches, including finerenone in DKD.
Reviewer 2 Report
Comments and Suggestions for Authors
Thank you for the opportunity to review this manuscript.
Piko et al provide a review of finerenone in the treatment of diabetic kidney disease. They provide an overview of diabetic kidney disease, brief overview of steroidal MRAs then move onto a review of non-steroidal MRAs including both pre-clinical and clinical data.
Comments:
1. Consider updating to the title of the manuscript to reflect that the focus of the review is on application to kidney disease, perhaps: "Finerenone: from the mechanism of action to clinical use in kidney disease"
2. In section 1, please provide a reference for the statement "Accounting for half of all cases, diabetes is the leading cause of ESKD worldwide."
3. Please add in a section to discuss limitations/drawbacks/risks of finerenone, what is currently unknown or still needs to be determined and some challenges that may arise in the implementation of finerenone in clinical practice (e.g. costs, use with other medications causing hyperkalaemia, etc).
4. Please include comment how finerenone could fit into the current therapeutic landscape for diabetic kidney disease.
5. In the conclusion, there is the statement "could soon
become an important new treatment option in non-diabetic CKD". There is limited mention of non-diabetic CKD in the main text - please provide more information regarding potential use in non-diabetic CKD and possible mechanisms of action.
Author Response
Reviewer #2
Thank you for the opportunity to review this manuscript.
Piko et al provide a review of finerenone in the treatment of diabetic kidney disease. They provide an overview of diabetic kidney disease, brief overview of steroidal MRAs then move onto a review of non-steroidal MRAs including both pre-clinical and clinical data.
Comments:
Consider updating to the title of the manuscript to reflect that the focus of the review is on application to kidney disease, perhaps: "Finerenone: from the mechanism of action to clinical use in kidney disease"
Response: Thank you for your comment. We have changed the title according to your suggestion.
In section 1, please provide a reference for the statement "Accounting for half of all cases, diabetes is the leading cause of ESKD worldwide."
Response: Thank you for your comment. We have added the reference.
- Please add in a section to discuss limitations/drawbacks/risks of finerenone, what is currently unknown or still needs to be determined and some challenges that may arise in the implementation of finerenone in clinical practice (e.g. costs, use with other medications causing hyperkalaemia, etc).
Response: Thank you for your valuable and interesting comment. We have added some new data on drug-drug interactions and also on the cost-effectiveness of finerenone.
- Please include comment how finerenone could fit into the current therapeutic landscape for diabetic kidney disease.
Response: Thank you for your comment. We have added an additional subheading regarding the treatment strategies for diabetic kidney disease, including finerenone.
- In the conclusion, there is the statement "could soon become an important new treatment option in non-diabetic CKD". There is limited mention of non-diabetic CKD in the main text - please provide more information regarding potential use in non-diabetic CKD and possible mechanisms of action.
Response: Thank you for your response. There is very little data on this topic. We have previously mentioned the potential of treatment Alport syndrome and other collagenopathies with finerenone. We have now added an additional, very recent study on this topic, as well.
Round 2
Reviewer 1 Report
Comments and Suggestions for Authors
According to my knowledge, the present form this review will be acceptable for the journal communications. But the conclusion needs to be revise moderately.
Comments on the Quality of English LanguageAccording to my knowledge, the present form this review will be acceptable for the journal communications. But the conclusion needs to be revise moderately.
Author Response
Thank you for your response and hard work on our manuscript.
We have changed the conclusion part of the manuscript.